# A Study of Differences in Enjoyment, Exercise Commitment, and Intention to Continue Participation Among Age Groups of Adult Amateur Golfers

**DOI:** 10.3390/bs15030398

**Published:** 2025-03-20

**Authors:** Hye Jin Yang, Ji-Hye Yang, Chulhwan Choi, Chul-Ho Bum

**Affiliations:** 1Department of Physical Education, Graduate School, Kyung Hee University, Seocheon-dong 1, Giheung-gu, Yongin-si 17104, Gyeonggi-do, Republic of Korea; y0108577@khu.ac.kr; 2Department of Human Movement Studies and Special Education, Old Dominion University, Norfolk, VA 23529, USA; jyang009@odu.edu; 3Department of Physical Education, Gachon University, 1342 Seongnamdaero, Sujeong-gu, Seongnam-si 13120, Gyeonggi-do, Republic of Korea; 4College of Physical Education, Kyung Hee University, Seocheon-dong 1, Giheung-gu, Yongin-si 17104, Gyeonggi-do, Republic of Korea

**Keywords:** exercise enjoyment, exercise satisfaction, golf, lifelong exercise, middle-aged adults, physical health, older adults, skill improvement, social relationship, stress relief

## Abstract

Golf is one of the leisure sports that offers various benefits and has become a popular public sport with participation from people of various age groups. A total of 262 questionnaires were distributed online, with 240 valid responses collected after excluding 22 with non-responses or partial answers. The results showed statistically significant differences in enjoyment by age group, except for recognition from others. Regarding physical health, those in their 60s and above had higher mean scores than other age groups, and those in their 50s scored higher than those in their 30s. Stress relief was greater among those in their 50s than among those in their 20s and 30s, while socialization was higher among those in their 50s than in their 20s. However, for skill improvement, participants in their 20s scored higher than those in their 50s and 60s and above. Exercise commitment and intention to continue participation also varied significantly by age group, with older participants generally scoring higher. In conclusion, the results of this study revealed significant differences in psychological factors among the age groups. Specifically, amateur golfers aged 50 and above showed higher mean scores in all variables compared to younger groups.

## 1. Introduction

Historically, golf has been a leisure sport played by people in certain income groups and age groups and can provide physical, mental, and social benefits ([30]). Golf is considered a technically challenging competitive sport that provides psychological fulfillment and satisfaction while also having psychological benefits, such as stress relief from playing in nature ([58]). Golf has historically not been a widely adopted sport due to its psychological inaccessibility and economic burden ([43]). However, since the restriction of indoor sports following COVID-19, the number of golfers in their 20s and 40s has increased, with various sports attracting new participants of different ages ([27]).

One reason for this shift is the change in consumption behavior. While utilitarian and savings-oriented consumption were once prominent, self-centered consumption emphasizing self-expression and experience has become a focal point ([35]). Particularly, consumption as a demonstration of self-worth has increased among those in their 20s and 30s ([37]), with many using social media to showcase their golf experiences, reinforcing its image as a “rich sport” ([10]; [68]). These consumption tendencies are linked to other psychological factors ([40]). Social difficulties such as exorbitant real estate prices and high unemployment rates in the Republic of Korea have caused significant stress ([1]); people in their 20s and 30s tend to relieve this stress through self-expressive consumption ([59]). These changes show that golf is no longer limited to a certain class or age group but is becoming a popular sport that various generations can enjoy.

These age differences have much to do with the fundamental motivation and enjoyment behind sports ([13]). Enjoyment of sports is a key factor in sustaining participation and is a persistent motivator ([23]). Enjoyment through participating in sports is a positive emotion, and such “enjoyment” from participating in sports ([55]) keeps people engaged in sports. Even when physical fitness is the primary motivation for sport participation, people are more likely to cease if they do not enjoy it ([21]). This enjoyment factor can be experienced differently by different age groups ([54]), with younger golfers favoring technical skill development and enjoying competitive challenges ([22]). Conversely, middle-aged and older golfers tend to favor factors such as social interaction and health benefits derived from golf rather than technical achievement ([30]; [51]). Understanding these enjoyment factors can improve engagement and continued participation in exercise.

Exercise commitment is a key factor in maximizing enjoyment and sustained participation in sports. Exercise commitment refers to how committed and engaged an individual is in a sporting activity ([17]; [47]). The state of commitment is related to both performance and enjoyment of a sport ([67]). Sports commitment is driven by high expectations for skill improvement or a competitive drive to compete ([60]). It can be experienced by anyone participating in sports, not just athletes ([61]). According to [28] ([28]), exercise immersion is critical to enhancing positive psychological states and increasing the intention to continue participating in golf.

Intention to continue participation is an important variable because, unlike initial exercise participation, ongoing engagement provides participants with numerous physical and mental benefits, as well as enjoyment ([16]). Intention to continue participation is defined as a psychological state in which participants intend to continue exercising in the future ([16]) or a commitment to a sport when they are fully engaged ([53]). [65] found that individuals who continue to participate in sports experience positive emotions.

Therefore, this study compared and analyzed the differences in enjoyment, exercise commitment, and intention to continue participation among adult amateur golfers by age. This study explores the psychological differences among age groups and provides a basis for improving the enjoyment and retention of adult amateur golfers. Furthermore, the results of this study contribute to the development of measures to revitalize golf participation across age groups. The hypothesis established in this study is as follows:

**Hypothesis** **1.** 
*There are differences in enjoyment, exercise commitment, and intention to continue participating among adult amateur golfers depending on age.*


## 2. Materials and Methods

### 2.1. Study Participants

This study was conducted on adult amateur golfers with prior experience in playing golf. The sampling method was non-probability convenience sampling. The sample consisted of members of golf driving ranges in the Gyeonggi-do, Seoul, and Gangwon-do provinces in the Republic of Korea. A total of 262 questionnaires were collected. After excluding 22 questionnaires with incomplete responses or missing data, 240 valid questionnaires were used for analysis. In the case of convenience sampling, there may be limitations in terms of generalizability and representativeness compared to probability sampling. However, since this study specifically targeted amateur golfers with prior golf experience, convenience sampling appeared to be a more suitable method. Moreover, as the response rate and sample size were sufficiently secured, the reliability of the study was ensured, which helps mitigate the limitations of convenience sampling. The socio-demographic characteristics of the participants are presented in Table 1.

### 2.2. Instruments

This study used a structured questionnaire as the measurement instrument. It consisted of seven questions on general characteristics (socio-demographic information), twenty on enjoyment, four on exercise commitment, and four on intention to continue participation. The enjoyment variable, developed by [44] ([44]) and used by [34] ([34]) to analyze golf enjoyment, was modified and adapted to fit the topic of this study. This instrument was used to measure the degree of enjoyment in individuals’ golf experience, with higher scores indicating a greater level of enjoyment. It comprised five subfactors (physical health, skill improvement, stress relief, recognition from others, and social relationships), with a total of 20 questions. The exercise commitment variable, developed by [29] ([29]) and adapted by [36] ([36]) in the Republic of Korea, was modified and adapted to fit this study. Exercise commitment measured the extent to which individuals fully concentrate on the activity and experience a sense of unity between body and mind, for which higher scores indicate a greater level of commitment. It is a single-factor scale with four items. Lastly, the intention to continue participating variable, used by [14] ([14]) and [33] ([33]), was also modified and adapted for this study. Intention to continue participation measured the willingness and desire to keep engaging in the sport, for which higher scores indicate a stronger intention to continue participation. It is a single-factor scale with four items. All scales used a 5-point Likert scale ranging from 1 (“not at all”) to 5 (“very much so”).

### 2.3. Data Analysis

SPSS software (version 28.0) was used for all data processing. First, a frequency analysis was conducted to determine socio-demographic characteristics. Second, Exploratory Factor Analysis was conducted to ensure the validity of the questionnaire. The Kaiser–Meyer–Olkin (KMO) and Bartlett’s tests of sphericity, principal component factor extraction, and orthogonal varimax were used for factor analysis. Subsequently, Cronbach’s α reliability coefficient was used to verify internal consistency. Third, correlation analysis was conducted to analyze the relationships between each factor. Finally, a multivariate analysis of variance (MANOVA) with post-hoc analysis was conducted to compare the differences in enjoyment, exercise commitment, and intention to continue participation among adult amateur golfers by age group.

## 3. Results

### 3.1. Scale Validity and Reliability

A validity analysis of the enjoyment factor showed that the KMO value was 0.859 and the Bartlett value was *p* < 0.001, which was suitable for exploratory factor analysis. Five subfactors were extracted: skill improvement, physical health, stress relief, social relationships, and recognition by others. The results of the reliability test showed that all factors exceeded the 0.700 threshold, with 0.927 for skill improvement, 0.939 for physical health, 0.926 for stress relief, 0.895 for social relationships, and 0.878 for the recognition of others, confirming the reliability of the instrument (Table 2).

A validity analysis of exercise commitment showed that the KMO value was 0.768 and the Bartlett value was *p* = 0.001, which was suitable for exploratory factor analysis. The reliability test showed that the value of 0.857 for exercise commitment exceeded the threshold of 0.700, thus confirming the instrument’s reliability (Table 3).

A validity analysis of the intention to continue participation showed that the KMO value was 0.816, and the Bartlett value was *p* = 0.001, which was suitable for exploratory factor analysis. The reliability test showeda value of 0.922, which exceeded the threshold of 0.700, thus confirming the instrument’s reliability (Table 4).

### 3.2. Correlation Analysis for Multicollinearity

Pearson’s correlation analysis was used to analyze the relationships between variables. The results showed that most variables were statistically significant. In detail, seven results exhibited moderate correlations of at least 0.40, with the highest result between stress relief and exercise commitment (*r* = 0.578). Additionally, the correlation coefficient was < 0.80, indicating that there was no multicollinearity (Table 5).

### 3.3. MANOVA on Dependent Variables by Age Groups

For comparative analysis, a MANOVA was conducted to test the differences in (a) enjoyment, (b) exercise commitment, and (c) intention to continue participation by the five age groups (Table 6). First, the homogeneity of covariance was tested (Box’s *M* = 205.351, *F* = 1.701, *p* < 0.05). Statistically significant differences were verified among the five groups (Wilks’ lambda = 0.652, *F* = 3.721, *p* < 0.05, partial *η*^2^ = 0.101). Statistically significant mean differences were found on six dependent variables except for the recognition from others sub-factor (*F* = 1.657, *p* > 0.05): (a) physical health (*F* = 8.423, *p* < 0.001), (b) skill improvement (*F* = 4.372, *p* < 0.01), (c) stress relief (*F* = 3.897, *p* < 0.01), (d) social relationship (*F* = 3.528, *p* < 0.01), (e) exercise commitment (*F* = 6.609, *p* < 0.001), and (f) intention to continue participation (*F* = 4.064, *p* < 0.01).

As this study analyzed the statistical differences among the five age groups, post-hoc analyses were conducted to determine the presence of significant group differences. For the physical health factor, post-hoc analysis showed that the 60+ age group had a higher mean difference than the other age groups; the 50s age group had a higher mean difference than the 30s. Regarding skill improvement, the 20s age group had a higher mean value than the 50s and 60s and above age groups. For the stress relief factor, those in their 50s scored higher than those in their 20s and 30s. In the social relationship factor, those in their 50s had higher mean values than those in their 20s. The post-hoc analysis of the exercise commitment factor showed that those in their 50s and 60s and above had higher mean values than those in their 20s and 30s. Additionally, those in their 40s had higher mean values than those in their 30s. Finally, in the post-hoc analysis of the intention to continue participation, those in their 60s and above had higher intentions than those in their 20s, 30s, and 40s. The detailed results of the post-hoc analyses and the mean scores of the dependent variables for the four groups are reported in Table 7 and Table 8.

## 4. Discussion

This study is a comparative analysis of enjoyment, exercise commitment, and intention to continue participating among adult amateur golfers of different age groups, aiming to help improve enjoyment and retention among amateur golfers. The results are discussed below.

### 4.1. Exercise (Golf) Enjoyment

Adult amateur golfers’ enjoyment of golf by age group differed significantly in all factors except recognition from others. Regarding physical health, the 60+ year age group had higher mean scores than the other age groups, and those in their 50s had higher mean scores than those in their 30s. This may be due to the gradual decline in physical health with age, but it may also be due to the characteristics of golf, which make it a suitable exercise for older adults. While continuous participation in leisure sports effectively promotes physical fitness and health ([31]), participation rates tend to decline with age as physical ability declines ([4]). However, golf requires minimal physical exertion and only short walks between rounds ([50]), making it an effective but low-impact sport that can be enjoyed at an older age ([62]). Recent research reports the positive mental and physical health benefits of golf ([5]), and [11] ([11]) found that those aged 40–60 reported higher enjoyment of the sport than those in their 20s and 30s. [57] ([57]) found that golfers over the age of 60 perceived golf as beneficial for their physical and mental well-being and found it enjoyable. Younger golfers typically lack health-related motivations for playing ([66]), whereas older golfers are more likely to appreciate its physical health benefits.

This study showed that participants in their 20s had higher average mean scores than those in their 50s, 60s, and older on skill improvement. While most factors in this study had higher mean scores for older age groups, those in their 20s had the highest scores for skill improvement. Skill improvement reflects a golfer’s “perceived ability”, including good scores and successful shots in a round ([63]), a key factor in sport enjoyment at any age ([7]). However, the higher scores among the 20s group in this study may be due to differing leisure priorities, with younger people tending to focus on the competitive aspect of the sport and older people focusing on the social aspect ([48]). [62] ([62]) found that the anticipation of great shots, excitement, and competitive pressure are important factors in participation, particularly among younger men, supporting this study’s findings. Furthermore, as older golfers experience physical and cognitive decline, their reluctance to perform skills or compete may decrease their interest in the sport ([25]).

The remaining two factors—social relationships and stress relief—were also favored among older adults, with those in their 50s scoring higher than those in their 20s for socializing. This aligns with the increasing value older adults place on social connections, consistent with [8] ([8])’s socioemotional selectivity theory. Their theory states that people who feel that their time is limited as they age value positive interpersonal relationships more than competition. [6] ([6]) found that amateur golfers over the age of 60 participated in golf for both health and social interaction. [39] ([39]) and [2] ([2]) reported that social connection and a sense of belonging are key motivators for golf participation. As people age, they are more likely to experience loneliness ([52]), which may be alleviated by golf participation.

Stress relief was also found to be higher among those in their 50s than among those in their 20s and 30s, likely due to golf’s social interactions and emotional bonding, which help reduce stress ([9]). Golf provides several opportunities for stress relief, such as socializing, escaping daily routines, and playing in nature ([62]). However, despite these benefits, [42] ([42]) and [41] ([41]) have shown that golfers are more likely to experience negative emotions when they lose a competition or poor performance. This may explain the relatively lower levels of stress relief among people in their 20s and 30s, as younger generations tend to enjoy the competition and challenging opportunities provided by golf. Additionally, golf requires a substantial time and effort investment to reach a certain level of skill ([49]), and younger players may struggle to consistently commit to the game due to time and financial constraints. This lack of substantial achievement can lead to negative emotions ([46]), explaining the relatively low levels of stress relief among those in their 20s and 30s.

This study found no significant age-group differences in the sub-factors of enjoyment and recognition from others. Attention and praise from others are factors in sport participation because they foster growth ([18]). Previous studies indicate that recognition from others is an important factor as it positively influences golf participants’ exercise commitment and persistence ([15]; [56]). However, the lack of difference across age groups may stem from golf’s independent nature, where personal achievement is prioritized regardless of age. Additionally, amateur golfers typically play for leisure rather than competition ([38]).

### 4.2. Exercise Commitment with Intention to Continue Participation

Exercise commitment was higher among middle-aged and older adults, with those in their 50s and 60s and above scoring higher than those in their 20s and 30s, and those in their 40s scoring higher than those in their 30s. Exercise commitment refers to the feeling of being immersed in a sport and creating a unified mind–body experience ([17]), which enhances enjoyment while performing exercise ([26]). Studies have shown that higher levels of exercise commitment are associated with higher levels of enjoyment and intention to continue participating ([19]; [32]; [62]).

This study found that those in their 60s and older who are the most committed to exercise have higher continuance intention than other age groups. This aligns with findings on exercise commitment, as golf requires significant time and financial investment to achieve a certain skill level. Older adults—having more time and financial resources—are more likely to be committed and continuously involved ([24]). [3] ([3]) showed that younger people tend to find the time-consuming nature of golf burdensome due to academic or work commitments. Additionally, the long duration of rounds and travel to courses, typically outside city centers, further discourage participation among younger players ([11]; [12]). The cost of the sport is also a factor; compared to other sports, golf is expensive in terms of equipment, club fees, and memberships. Younger people still perceive golf as a “rich sport” associated with high social status ([20]).

This study compared the differences in enjoyment, exercise commitment, and intention to continue participating among adult amateur golfers by age group and discussed the reasons for these differences. Psychological differences were found across age groups, with middle-aged and older adults deriving more enjoyment from golf for health, stress relief, and socialization; they also had greater exercise commitment and a higher propensity to continue participating. Younger adults derive more enjoyment from the competitive elements of the sport, such as improving their scores. These findings show that social environments and life goals change with age, thereby affecting golf participation.

Thus, it is necessary to use the results of this study as foundational data to provide age-appropriate programs and environments in order to establish golf as a ‘lifetime sport’ for all generations.

Based on the results of the middle-aged and elderly, high mean scores of all variables, except skill improvement, were found. It can be considered that golf is a popular sport for them, and they will continuously participate in golf. Physical health, stress relief, and social relationships were identified as the main motivating factors for playing golf, while satisfaction with skill improvement was lower compared to the younger group. This suggests that high levels of competition may contribute to stress among middle-aged and older adults and should be carefully considered.

On the other hand, although the influx of the younger generation in playing golf has increased compared to the past, the mean scores of enjoyment (except skill improvement), exercise commitment, and intention to continue participation were lower than those of the older generation. The results show that the participation of the younger generation in playing golf may decrease in the future. Thus, their participation motivation needs to be enhanced through new approaches.

Regarding these results, ‘Top Golf’, which more than 20 million people use per year, is not only an image of just golf practice, but is also a complex cultural space which includes entertainment, restaurants, and pubs. Specifically, it played a positive role in promoting the young generation’s participation through combining playing golf and diverse entertainment factors ([45]). As indicated by the results of this study, it is suggested that the younger generation could find fulfillment in elements such as competition, like the improvement of their skill.

To release time and financial burdens, the ‘Play 9’ campaign could be introduced, which involves playing only 9 holes instead of 18 ([64]), as implemented by the United States Golf Association (USGA), or the revised rule of play to shorten game time could be utilized, which will be good solutions. In conclusion, if these improvements are implemented, golf is expected to strengthen its position as a lifetime sport that can be enjoyed by all age groups.

### 4.3. Study Limitations and Further Research

The results of this study provide valuable insights into the differences in enjoyment, exercise commitment, and intention to continue participation among age groups of adult amateur golfers. However, there are some limitations, and the following suggestions are made for future research. First, while 240 questionnaires were administered in this study, a larger and more diverse sample is needed to generalize the findings. Thus, future studies should aim to collect a larger sample size. Second, this study only focused on positive psychological factors such as enjoyment, exercise commitment, and intention to continue participation among adult amateur golfers. Future studies should also consider negative aspects such as sport dropout, stress, and dissatisfaction, as these factors are important for identifying differences between age groups.

## 5. Conclusions

This study analyzed the differences in enjoyment, exercise commitment, and intention to continue participation among adult amateur golfers by age group. First, enjoyment showed statistically significant differences across age groups for all factors except recognition from others. Regarding physical health, those in their 60s and older scored higher than those in other age groups, while those in their 50s showed a higher mean score than those in their 30s. The mean stress relief factor score was higher in those in their 50s than in those in their 20s and 30s; social relationships were also higher in those in their 50s than in those in their 20s. However, participants in their 20s had higher scores for skill improvement than those in their 50s and 60s and above. Second, there were significant differences in exercise commitment by age group. Those in their 50s and 60s and older were more committed than were those in their 20s and 30s; those in their 40s were more committed than were those in their 30s. Third, the intention to continue participation was also significant according to age group. Those in their 60s and older scored higher than those in their 20s, 30s, and 40s.

The results showed that the 40+ age group had slightly higher mean scores for enjoyment, exercise commitment, and intention to continue playing golf, indicating that golf has many mental, physical, and social benefits in this age group. In contrast, the younger generations had relatively lower mean scores across all factors, indicating that they were influenced by their social environment and personality traits.

## Figures and Tables

**Table 1 behavsci-15-00398-t001:** Socio-demographic information of study participants by age group.

		20s—a	30s—b	40s—c	50s—d	60s and Above—e
Gender	Male	21 (50.0%)	36 (63.2%)	21 (48.8%)	46 (64.8%)	15 (55.6%)
	Female	21 (50.0%)	21 (36.8%)	22 (51.2%)	25 (35.2%)	12 (44.4%)
Golf experience	Less than 1 yrs	17 (40.5%)	15 (26.3%)	5 (11.6%)	1 (1.4%)	1 (3.7%)
	1—less than 3 yrs	13 (31.0%)	21 (36.8%)	6 (14.0%)	5 (7.0%)	2 (7.4%)
	3—less than 5 yrs	9 (21.4%)	16 (28.1%)	15 (34.9%)	15 (21.1%)	9 (33.3%)
	5—less than 10 yrs	2 (4.8%)	5 (8.8%)	13 (30.2%)	23 (32.4%)	9 (33.3%)
	10—less than 20 yrs	1 (2.4%)	-	3 (7.0%)	20 (28.2%)	2 (7.4%)
	More than 20 yrs	-	-	1 (2.3%)	7 (9.9%)	4 (14.8%)
Mastery (golf handicap)	Less than 10	8 (19.0%)	14 (24.6%)	8 (18.6%)	9 (12.7%)	3 (11.1%)
	10—less than 15	6 (14.3%)	9 (15.8%)	3 (7.0%)	19 (26.8%)	8 (29.6%)
	15—less than 20	1 (2.4%)	6 (10.5%)	13 (30.2%)	22 (31.0%)	3 (11.1%)
	More than 20	12 (28.6%)	10 (17.5%)	13 (30.2%)	19 (26.8%)	9 (33.3%)
	I do not know	15 (35.7%)	18 (31.6%)	6 (14.0%)	2 (2.8%)	4 (14.8%)
Frequency of golf practice	None	5 (11.9%)	9 (15.8%)	4 (9.3%)	8 (11.3%)	1 (3.7%)
	1–2 times a month	19 (45.2%)	9 (15.8%)	7 (16.3%)	20 (28.2%)	4 (14.8%)
	2–3 times a week	8 (19.0%)	20 (35.1%)	16 (37.2%)	26 (36.6%)	8 (29.6%)
	3–4 times a week	8 (19.0%)	14 (24.6%)	10 (23.3%)	15 (21.1%)	13 (48.1%)
	Almost everyday	2 (4.8%)	5 (8.8%)	6 (14.0%)	2 (2.8%)	1 (3.7%)
Frequency of playing golf	None	10 (23.8%)	17 (29.8%)	5 (11.6%)	3 (4.2%)	4 (14.8%)
	1–5 times a year	6 (14.3%)	2 (3.5%)	2 (4.7%)	1 (1.4%)	1 (3.7%)
	1–2 times a month	18 (42.9%)	29 (50.9%)	31 (72.1%)	47 (66.2%)	17 (63.0%)
	3–4 times a month	6 (14.3%)	4 (7.0%)	3 (7.0%)	14 (19.7%)	2 (7.4%)
	5–6 times a month	1 (2.4%)	4 (7.0%)	1 (2.3%)	3 (4.2%)	1 (3.7%)
	More than 7 times	1 (2.4%)	1 (1.8%)	1 (2.3%)	3 (4.2%)	2 (7.4%)
	a month					
	Total	42 (100%)	57 (100%)	43 (100%)	71 (100%)	27 (100%)

Note. a = 20s, b = 30s, c = 40s, d = 50s, e = 60s and Above.

**Table 2 behavsci-15-00398-t002:** Results of factor analysis for enjoyment factor.

Items	1	2	3	4	5
SI—1: I feel good when I play better than others during a match.	**0.922**	−0.038	0.104	0.085	0.125
SI—2: I feel thrilled when I hit the ball well.	**0.907**	−0.066	0.073	0.106	−0.009
SI—3: I feel proud when I achieve a better record than usual.	**0.884**	−0.052	0.107	0.151	−0.015
SI—4: I enjoy improving my golf skills.	**0.837**	−0.074	0.122	0.087	0.180
PH—1: I feel that playing golf helps improve my health.	−0.081	**0.901**	0.084	0.158	0.227
PH—2: I feel that my physical strength gradually improves while playing golf.	−0.108	**0.887**	0.113	0.173	0.221
PH—3: I feel that my physical strength improves when I play golf.	−0.105	**0.882**	0.139	0.206	0.095
PH—4: I feel that my physical condition improves when I play golf.	0.030	**0.861**	0.123	0.060	0.100
STR—1: I feel relieved from complex problems when I play golf.	0.133	0.093	**0.915**	0.125	0.097
STR—2: I feel free from daily life stress when I play golf.	0.046	0.073	**0.886**	0.151	0.025
STR—3: I feel happy when I play golf.	0.089	0.182	**0.849**	0.168	0.119
STR—4: I feel mentally relaxed when I play golf.	0.163	0.111	**0.835**	0.222	0.093
SOC—1: I like enjoying golf with acquaintances.	167	0.107	139	**0.859**	0.063
SOC—2: Golf allows me to socialize freely.	0.105	0.136	0.191	**0.844**	0.198
SOC—3: I play golf to spend time with friends.	0.232	0.155	0.168	**0.822**	0.046
SOC—4: Golf increases my opportunities for social interaction.	−0.025	0.180	0.177	**0.802**	0.138
REC—1: I feel recognized by others when I play well.	0.174	0.158	−0.031	0.117	**0.861**
REC—2: I like being seen as good at golf.	0.073	0.156	0.103	0.184	**0.846**
REC—3: I feel socially elevated when I play well.	−0.125	0.091	0.082	0.024	**0.819**
REC—4: I like receiving positive evaluations from others when I play golf.	0.220	0.218	0.188	0.112	**0.790**
Eigenvalues	6.627	3.704	2.409	1.884	1.604
Variance (%)	33.133	18.520	12.046	9.421	8.018
Cronbach’s alpha	0.927	0.939	0.926	0.895	0.878

Note. KMO = 0.859, χ^2^ = 4035.406, df = 190, sig < 0.001, SI = skill improvement, PH = physical health, STR = stress relief, SOC = social relationship, REC = recognition from others.

**Table 3 behavsci-15-00398-t003:** Results of the factor analysis for exercise commitment factor.

Items	Factor 1
EC—1: I invest a lot of time and effort in golf.	0.897
EC—2: I prioritize reading newspapers, magazines, and watching TV broadcasts about golf.	0.869
EC—3: I put a lot of effort into improving my game.	0.811
EC—4: I feel highly focused when I play golf.	0.788
Eigenvalues	2.839
Variance (%)	70.963
Cronbach’s alpha	0.857

Note. KMO = 0.768, χ^2^ = 479.786, df = 6, sig = 0.001, EC = exercise commitment.

**Table 4 behavsci-15-00398-t004:** Results of the factor analysis for intention to continuous participation.

Items	Factor 1
ICP—1: I intend to continue playing golf.	0.941
ICP—2: I will not quit playing golf.	0.931
ICP—3: I will continue playing golf even if I face uncomfortable situations.	0.902
ICP—4: I will continue to play golf whenever I have time.	0.861
Eigenvalues	3.308
Variance (%)	82.294
Cronbach’s alpha	0.922

Note. KMO = 0.816, χ^2^ = 773.330, df = 6, sig = 0.001, ICP = intention to continue participation.

**Table 5 behavsci-15-00398-t005:** Results of the correlation analysis on dependent variables.

	1	2	3	4	5	6	7
Physical health	1						
Skill improvement	−0.083	1					
Stress relief	0.279 **	0.244 **	1				
Recognition from others	0.366 **	0.175 **	0.240 **	1			
Social relationship	0.353 **	0.259 **	0.408 **	0.303 **	1		
Exercise commitment	0.286 **	0.117	0.578 **	0.434 **	0.410 **	1	
Continuous participation intent	0.322 **	0.220 **	0.492 **	0.367 **	0.540 **	0.538 **	1

Note. ** *p* < 0.01.

**Table 6 behavsci-15-00398-t006:** Results of the MANOVA.

Variables	Sub-Factors	*df*	*F*	*p*	*η* ^2^	Tukey HSD
Enjoyment	Physical health	4	8.423	<0.001 ***	0.125	a, b, c, d < e/b < d
	Skill improvement	4	4.372	0.002 **	0.069	a > d, e
	Stress relief	4	3.897	0.004 **	0.062	a, b < d
	Recognition from others	4	1.657	0.161	0.027	-
	Social relationship	4	3.528	0.008 **	0.057	a < d
Exercise commitment		4	6.609	<0.001 ***	0.101	a, b < d, e/b < c
Intention to continue participation		3	4.064	0.003 **	0.065	a, b, c < e

Note. *** *p* < 0.001, ** *p* < 0.01; a = 20s; b = 30s; c = 40s; d = 50s; e = 60s and above.

**Table 7 behavsci-15-00398-t007:** Mean scores of the dependent variables by each group.

		Enjoyment				Exercise Commitment	Intention to Continuous Participation
	1	2	3	4	5		
20s	3.208	4.482	3.488	3.381	3.667	3.321	3.964
30s	2.877	4.364	3.517	3.250	3.759	3.241	3.882
40s	2.965	4.064	3.942	3.424	3.808	3.686	3.957
50s	3.430	4.032	3.951	3.574	4.099	3.792	4.116
60s and above	4.018	3.991	3.917	3.648	4.167	3.926	4.528

Note. 1 = physical health, 2 = skill improvement, 3 = stress relief, 4 = recognition from others, 5 = social relationships.

**Table 8 behavsci-15-00398-t008:** Results of the post-hoc analyses from five age groups.

			Enjoyment				Exercise Commitment	Intention to Continuous Participation
		1	2	3	4	5		
20s	G2	0.415	0.925	1	0.939	0.977	0.987	0.981
	G3	0.775	0.055	0.1	0.999	0.918	0.205	1
	G4	0.745	0.011 *	0.042 *	0.761	0.037 *	0.019 *	0.825
	G5	0.005 **	0.043 *	0.243	0.696	0.072	0.017 *	0.018 *
30s	G1	0.415	0.925	1	0.939	0.977	0.987	0.981
	G3	0.99	0.228	0.097	0.841	0.998	0.043 *	0.986
	G4	0.010 *	0.069	0.035 *	0.193	0.103	<0.001 ***	0.378
	G5	<0.001 ***	0.166	0.258	0.253	0.166	0.002 **	0.002 **
40s	G1	0.755	0.055	0.1	0.999	0.918	0.205	1
	G2	0.99	0.228	0.097	0.841	0.998	0.043 *	0.986
	G4	0.081	0.999	1	0.888	0.301	0.956	0.796
	G5	<0.001 ***	0.994	1	0.813	0.331	0.724	0.015 *
50s	G1	0.745	0.011 *	0.042 *	0.761	0.037 *	0.019 *	0.825
	G2	0.010 *	0.069	0.035 *	0.193	0.103	<0.001 ***	0.378
	G3	0.081	0.999	1	0.888	0.301	0.956	0.796
	G5	0.047 *	0.999	1	0.995	0.995	0.943	0.099
60s and above	G1	0.005 **	0.043 *	0.243	0.696	0.072	0.017 *	0.018 *
	G2	<0.001 ***	0.166	0.258	0.253	0.166	0.002 **	0.002 **
	G3	<0.001 ***	0.994	1	0.813	0.331	0.724	0.015 *
	G4	0.047 *	0.999	1	0.995	0.995	0.943	0.099

Note. *** *p* < 0.001, ** *p* < 0.01, * *p* < 0.05; G1 = 20s; G2 = 30s; G3 = 40s; G4 = 50s; G5 = 60s and above; 1 = physical health; 2 = skill improvement; 3 = stress relief; 4 = recognition from others; 5 = social relationship.

## Data Availability

The data reported in this study are available upon request from the corresponding authors.

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
