# Peer review of "A Study of Differences in Enjoyment, Exercise Commitment, and Intention to Continue Participation Among Age Groups of Adult Amateur Golfers"

_behavsci, 2025, doi:10.3390/bs15030398_

Round 1
Reviewer 1 Report
Comments and Suggestions for Authors
The manuscript effectively explores important psychological factors influencing golfers' experiences across different age groups. Overall, the study is well-designed, with suitable statistical methods applied clearly and effectively. The authors have successfully highlighted significant age-related variations, which enrich our understanding of the psychological dimensions of sports participation and commitment.
However, the manuscript would benefit from briefly addressing potential limitations associated with the convenience sampling approach, including considerations about generalizability and representativeness of the sample.
Most importantly, the final chapter should be significantly expanded to address the critical "so what?" questions. It currently provides a basic summary, but elaborating on practical implications for sports practitioners, policy makers, and organizations would greatly enhance the manuscript’s relevance and applicability. Specifically, offering concrete recommendations on how these findings can inform targeted interventions or programs to promote sustained golf participation across different age groups, particularly younger players, would substantially strengthen the study's contribution.
Reviewer 2 Report
Comments and Suggestions for Authors
Dear authors,
Thank you for submitting this interesting article. Let me positively highlight the idea to study the differences in enjoyment, exercise commitment, and intention to continue participation among age groups of adult amateur golfers. In general, my opinion is that some more important, as well as, some minor “technical” issues of the manuscript should be reconsidered and resolved in a detailed way.
General comments
The abstract is under 200 words as requested. The abstract is a very, very important part of every study because it provides readers with a "window" into the study, so I suggest you consider the following: - insert a "background" sentence at the beginning of the abstract to motivate readers to read the abstract (and then the entire article), and you have enough material for that in the Introduction section; - consider whether to leave the current first sentence of the abstract in the paper, which is almost identical to the title of this article; - consider whether to reject this sentence in the abstract related to the methods: "Analysis, exploratory factor analysis, reliability testing, correlation analysis, and multivariate analysis of variance were conducted using SPSS 28.0." In my opinion, the previous sentence, which says enough about the methods, is just enough for abstract..; - consider adding a possible "conclusion" sentence at the end of the abstract: e.g., something like "how this study confirmed the value of golf for adult amateur players, especially for older adults (for 50s or more adults)" or something like your conclusion sentence "Psychological differences were found across age groups, with middle-aged and older adults deriving more enjoyment from golf for health, stress relief, and socialization; they also had greater exercise commitment and a higher propensity to continue participating". As well, please, consider changing the Keywords because they should not be identical to the words in the study title. You can also increase their number to up to 10 keywords: e.g., exercise enjoyment; exercise satisfaction, golf, lifelong exercise, middle-aged adults, physical health, older adults, skill improvement, social relationship, stress relief. Keywords, together with the words used in the study title, expand the possible reach and searching results of similar articles.
The Introduction section is very well written and detailed, with extensive literature cited. The Introduction is a very good part of the text at the beginning of this study. The citations in the text are well done.
Methods are written in a quality, concise and understandable way. In my opinion, the 2.2 Instruments paragraph should be expanded a bit with a better description of the instruments used. The measures used are a very important part of this study, and you described each measure with only two sentences. Also, please list the precise age groups (20s – a; 30s – b; 40s – c; 50s – d; 60s – e) at the top of the table in Table 1 instead of the numbered names as Groups 1,2,3,4,5. Note: later in the MANOVA table, using of the labels for the established differences for age groups is: a,b,c,d,e. For easier reading, the labels of age groups should be unique throughout the study.
The Results are generally well written and the tables are well organized and designed. I recommend that you list the items with each subfactor in Table 1, this could contribute to a better understanding and monitoring of the study. For example, format it as: SI-1 whole item (for Skill improvement); PH-2 whole item (for Physical health); STR-3 whole item (for Stress relief); SOC-1 whole item (for Social relationship); REC-3 whole item (for Recognition from others). I make a similar recommendation for the items in Tables 3 (for EC-1) and 4 (for ICP-3). I recommend that you interpret the results of Table 5 very briefly, at least the correlations above .40 (moderate correlations). Please consider swapping the places of Table 8 with Table 7 because it seems to me that it would be nicer to see the Mean scores of the variables immediately after the display of the MANOVA results.
Discussion – The discussion is written very well and based on the results of this research. However, I recommend that you consider dividing the discussion into subsections for easier reading and understanding and easier interpretation of the study's findings (like the one already made in the Results section). The study has two simple, meaningful and separate parts of the measures used: 1 – Exercise (golf) enjoyment, and 2 - Exercise Commitment with Intention to Continue Participation. At the end of the Discussion section, you can include a short Study limitations/further research paragraph. Recommendations for the implementation of future research arise partially from the author’s awareness about present study limitations.
Conclusion – This section is written very well and the conclusions are based on the findings of this research. It is possible to transfer the last part of the section (further research) to the last and recommended paragraph of the Discussion (Study limitations/further research paragraph).
Author Contributions: Please, according to BehavSci's Instructions for authors, use this form: „lH.J.Y., J.-H.Y., C.C. and C.-H.B. contributed to the writing of the published version of this manuscript.“ or look for the MDPI Credit taxonomy for other possible solutions.
References were made in accordance with BehavSci's Instructions for authors.
In general, I am of the opinion that this article is very worthy of publication after the open questions in this review are considered and resolved.
Specific comments
- Line 105 – Consider changing the word factor to the word measure (or variable or scale)
- Line 105 – Consider changing the formatting of the enjoyment factors - either putting them in Italic formatting or listing them as the title of the questionnaire (Enjoyment Factor or in Golf Enjoyment Scale)
- Lines 108-109 Consider changing the formatting of the exercise commitment scale – either putting them in Italic formatting or listing them as the title of the questionnaire (Exercise Commitment Scale)
- Line 111 – Consider changing the formatting of the intention to continue participating factor to Italic formatting
- Lines 125-128 - Consider moving this paragraph to Data analysis – so as not to repeat information.
- Line 136 – Table 2 - IMPORTANT! Please check factor saturations for Stress relief items!? It looks that they are copied from the field below (from Social relationship items) because they are identical. Check the whole table, please, as well.
- Line 154 – Tittle of this subsection should be numbered as 3.3., not 3.2. MANOVA on Dependent Variables by Age Groups
- Line 165 – What post-hoc test was used (Fisher’s LSD, Scheffe, Bonferoni, Tukey, ...)? Please specify it, either here or in Data Analysis paragraph.
- Line 184 - Table 8 - Please shorten the results to three decimal places after the comma. In my opinion, all results in tables except for coefficients of significance of differences (e.g., p<.001 or p=.048) or effect size coefficients (e.g., using CFA, …) should be shown to only two decimal places because it is easier to follow the study. Using a third decimal place does not contribute much, and makes it more difficult to follow (e.g., using of 3.537 or 3.54). However, I am aware that SPSS outputs its output with 3 decimal places, so I consider its use justified.
- Line 425 – Provide the page numbers of the Scanlan & Simmons (1992) chapter.
Reviewer 3 Report
Comments and Suggestions for Authors
Dear authors,
thank you for providing us with an interesting view to golf. After reading your paper, allow me to make some suggestions for its improvement:
- in the chapter 2.2. Study participants you explain that participants were amateur golfers aged 20 years or older who regularly participated in golf for over one year. Yet, in Table 1. you show us that there are some participants that with golf experience less than 1 year, with none frequency of golf practice or golf playing, which seems to be rather contradictory - please elaborate and explain.
- in Table 1. we encounter Group 1, 2, 3, 4 and 5 with no explanations what are these groups. We can assume that these are age groups - this is explained later on at the bottom of Table 8., but it should be explained here.
- I find it very unusual to present items connected with each enjoyment factor, exercise commitment and intention to continue participation to be shown as 1, 2, 3 and 4. Is there a reason as to why the items can not be shown? I should emphasize that not showing the items makes it harder to give conclusions because we might not be familiar with what was tested in those factors.
- the previous point is connected with the fact that some of the given conclusions are not supported by the research. For example, you mention that participation among adults in their 20s and 30s has risen significantly, but later on give recommendations how to boost participation of younger players. In this sense you indicate the problem of barriers to entry (assuming you mean money and time) and need to develop age-specific programs - was this also the part of your research, or is this conclusion based on studied literature? Please elaborate on your conclusion that younger generations were influenced by their social environment and personality traits.
Round 2
Reviewer 1 Report
Comments and Suggestions for Authors
The authors addressed my comments well. Good job on the revision.
Comments on the Quality of English LanguageGood.